# Measuring four facets of emotion beliefs in Germany: A German-language adaptation of the EBQ and its comparability across gender and different emotion abilities

**Raphael Gutzweiler**[1]*, **David J. Grüning**[2,3]

**1** University of Kaiserslautern-Landau (RPTU), Landau, Germany, **2** GESIS - Leibniz Institute for the Social Sciences, Department of Survey Design & Methodology, Mannheim, Germany, **3** Department of Psychology, Heidelberg University, Heidelberg, Germany

* raphael.gutzweiler@rptu.de

**Data Availability Statement:** The data for this study are not publicly available due to European data policy restrictions. However, interested researchers may contact the Local Ethics

## Abstract

Adaptive emotion regulation, involving the modulation of positive and negative emotions based on goals, is a crucial function for a person's mental health and general well-being. Factors influencing successful emotion regulation include beliefs about emotions, such as the controllability and usefulness of emotions. The Emotion Beliefs Questionnaire (EBQ) was developed to assess these beliefs and has shown promise in predicting emotion regulation and psychopathology across different countries. This study aims to advance EBQ's generalizability in measuring emotion beliefs by examining the scale's different validities for a developed German version. In a German sample of 348 respondents, we show the scale's factorial and broader construct validity as well as its factors' reliability. Notably, we demonstrate that the German EBQ is mostly strictly measurement invariant across central sociodemographic variables like age and gender. Interestingly, we also find the scale to be robust across different levels of other psychological constructs such as emotional reactivity and efficacy.

## Introduction

Emotions are a fundamental aspect of human experience, influencing how individuals perceive and interact with their surroundings [1]. Considering that emotions are a constant aspect of human life, it is evident that individual hold beliefs about their origins, purpose and regulation [2]. These emotion beliefs are thought to shape how individuals understand and manage their emotions, with emerging research suggesting they may serve as a key mechanism in emotion regulation [3]. Given that difficulties in emotion regulation are linked to the development and maintenance of psychopathology [4], an understanding of how emotion beliefs contribute to these regulatory processes could offer valuable insights for improving mental health outcomes.

Committee of the Department of Psychology (lek@uni-landau.de; https://psy.rptu.de/forschung/lokale-ethikkommission-lek) for data requests. The Ethics Committee is responsible for reviewing these requests and ensuring they comply with relevant guidelines before any data can be shared.

**Funding:** The author(s) received no specific funding for this work.

**Competing interests:** The authors have declared that no competing interests exist.

## Theoretical framework

Adaptive emotion regulation, defined as the up- and down-regulation of positive and negative emotions depending on regulatory goals [5], is associated with better mental health [4, 6–8]. To promote better mental health, it is important to examine which factors lead to more successful emotion regulation. According to Gross' extended process model [1], emotion regulation consists of four stages: identification, selection, implementation, and monitoring. During the identification stage, an individual decides whether to regulate the identified emotion, during the selection stage, an individual decides on the regulation strategy, during the implementation stage, an individual implements the previous chosen emotion regulation strategy, and during the monitoring stage, an individual evaluates the successful implementation. Multiple deficiencies at each stage can lead to maladaptive or unsuccessful emotion regulation. One factor are beliefs about emotions [9–13]: Superordinate beliefs about emotions (good versus bad; controllable versus uncontrollable) influence subsequent emotion regulation [14] by having an impact on effort and performance at all stages of Gross' process model of emotion regulation [5].

While there has been a growing number of research on the impact of malleability beliefs of emotions on emotion regulation (whether emotions are controllable, [9, 11, 13, 15–18], only few studies examined additional beliefs about the utility of emotions [19–22], its links to psychopathology [12, 23–25], or even emotion-specific control beliefs [26]. In general, beliefs about the (un-)controllability of emotions were associated with social anxiety [11, 17], schizophrenia-spectrum [23], psychological stress [24], depressive symptoms [27], and poorer mental health [28].

The growing body of research on beliefs about emotions was associated with multiple different terms to describe individuals' assumptions about their own emotions or emotions in general: implicit theories of emotions, emotion mindsets, and emotion beliefs [29]. All current labels have in common that they refer to how people explicitly state their perspectives on emotion (regulation) in self-report measures. At the same time, it has been found that differences exist between general emotion beliefs and personal emotion beliefs, with personal emotion beliefs having a higher impact on subsequent emotion regulation than general beliefs [10]. As Kneeland and Kisley [29] stated, assessing multiple emotion perspectives within the same sample will help clarify the redundancy and differentiation between the constructs of implicit theories (see ITES) and beliefs (see EBQ). This will go beyond Becerra et al.'s [19] approach comparison existing measures of beliefs about emotions.

## Emotion Beliefs Questionnaire

Presenting the Emotion Beliefs Questionnaire (EBQ), Becerra et al. [19] stated three goals which laid the foundations of their approach: A questionnaire assessing beliefs about emotion should assess controllability and usefulness of emotions separately, should assess beliefs about emotions in general, and provide valence-specific assessment of beliefs about controllability and usefulness of both positive and negative emotions. The authors argued that only with considering all three criteria, beliefs bout emotions could be assessed properly [19] and thus exceed the assessment of emotion beliefs with existing measures such as the Implicit Theories of Emotions Scale [13], Beliefs about Emotions Scale [30] or the Attitudes Toward Emotions Scale [31].

The EBQ consists of 16 items which were developed based on the theoretical foundations by Ford & Gross [2, 14]. The EBQ includes four theoretical factors on the controllability and usefulness dimensions across positive and negative emotions. However, in its initial validation in an online sample of 161 middle-aged Australian adults, confirmatory factor analyses

resulted in three instead of the assumed four factors: General-Controllability, Negative-Usefulness, and Positive-Usefulness. Fit indices showed moderate fit and internal consistency reliability was good [19]. Male participants showed on average more maladaptive beliefs about emotions than female participants. Inspecting the construct validity, measures included the Implicit Theories of Emotions Scale (ITES), the Emotion and Regulation Beliefs Scale (ERBS), the Beliefs about Emotions Scale (BES), the Perth Emotion Regulation Competency Inventory (PERCI), and the Depression Anxiety Stress Scales-21 (DASS-21) [19].

In accordance with the assumptions, the ITES was moderately linked to the General-Controllability subscale, whereas both usefulness subscales were not associated with the ITES. Further, all EBQ subscale and composite scores were significant predictors of most of the PERCI total scores [19], with the subscale General-Controllability being the strongest predictor of both positive- and negative-emotion regulation.

Examining its links with markers of psychopathology, the EBQ total score showed strong associations with all DASS-21 subscales. On the EBQ subscale level, only the EBQ subscale General-Controllability significantly predicted DASS-21 depression and stress subscale, but not the anxiety subscale [19]. Additionally, EBQ scores showed incremental predictive value above that of the ITES when predicting psychopathology and emotion regulation.

Since its first presentation and validation, the EBQ turned out to be popular and has since been validated in an Iranian and a US sample of adolescents and adults [21], an Italian sample of adults [22], two Japanese samples of university students [32], and a German sample of adults [20]. We will first give an overview over these studies, before we lay out why another validation study is needed.

In a sample of 104 German psychotherapists (young adults, predominantly female sample) in training, Biel et al. [20] used a different translation than the here presented translation and replicated the three factors found by Becerra et al. [19]: General-Controllability, Negative-Usefulness, and Positive-Usefulness. They applied an exploratory factor analysis trying to replicate the three-factor structure, thus not receiving fit indices. The resulting three factors showed acceptable internal consistencies. The EBQ total score was significantly associated with all DASS-subscales as measure of psychopathology. Further, higher EBQ total scores were linked with less emotional acceptance. Complementing the limitations stated by the authors, namely the small sample size and the unbalanced male-female ratio, both the missing fit indices and the limited range of measures to assess construct validity restricts interpretability and drawing conclusions for further studies. Applying their translation of the EBQ in a prospective study, beliefs about emotions (EBQ) predicted psychological stress related to somatic symptoms two weeks later over previous psychological stress in three samples [33], thus stressing the EBQ's predictive validity.

In three large samples of Iranian adolescents, Iranian adults, and American adults, Ranjbar et al. [21] examined the factor structure of the EBQ applying confirmatory factor analyses. Comparing multiple models, the 4-factor-model (Negative-Controllability, Positive-Controllability, Negative- Usefulness, Positive-Usefulness) had the best fit with the data within all three samples. Internal consistency reliability was moderate, with the highest scores in the subsample of American adults. Retest-reliability was reported to be moderate across all samples and scores. Construct validity was assessed using the ITES, PERCI, and DASS-21. Consistent with Becerra et al. [19], higher EBQ scores were moderately associated with higher ITES scores. However, in contrast to Becerra et al. [19], the EBQ usefulness subscales correlated significantly with ITES scores, even though less than the controllability subscales. Also, higher beliefs in the controllability and usefulness of emotions were associated with better emotion regulation abilities, assessed with the PERCI. Finally, greater overall maladaptive beliefs about

emotions correlated significantly with higher levels of depression, anxiety, and stress, assessed with DASS-21.

In 516 Italian adults, Rogier et al. [22] examined the construct validity of the EBQ using measures for emotion dysregulation (DERS, DERS-Positive) and psychopathology (DASS-21). Applying confirmatory factor analysis, the 4-factor (Negative-Controllability, Positive-Controllability, Negative- Usefulness, Positive-Usefulness) model fitted best [22] with good internal consistency reliability. Full measurement invariance was reached for gender, however, the authors did not examine mean differences in their sample. Comparable to findings of Becerra et al. [19], the EBQ controllability subscales were most predictive: Less beliefs about controllability of emotions was associated with higher difficulties emotion regulation, assessed with the DERS. Associations between DASS-21 and EBQ were found for all subscales except for the link between the EBQ Positive-Usefulness subscale and the DASS-21 Stress subscale [22]. Apart from administering the EBQ and DASS-21, Rogier et al. [22] did not use the same measures used by Becerra et al. [19], thus limiting results about the concurrent validity.

The previous validations have supported the EBQ's construct and concurrent validity. As the German validation lacked a comprehensive view on the EBQ's links with both measures of emotion (dys-)regulation, beliefs about emotions, and psychopathology in a large sample, the aim of the below-depicted study was to examine (a) the factorial structure and (b) the construct validity of the German EBQ in a large sample of young adults. By using multiple measures to validate the EBQ, we show a broader picture of the EBQ and its abilities to measure beliefs about emotions. In exploratory analyses, the objective is (c) to identify any differences in emotional beliefs as a function of gender, emotional reactivity, and emotional self-efficacy.

## Method

### Procedure

The sample was recruited through a convenience sampling technique. A digital survey was created using the SoSciSurvey platform and promoted through university mailing lists. Participants were encouraged to recruit other participants by receiving course credit. The survey took place in the period from 9 June to 2 September 2023. After an initial presentation of the study's aims and procedure as well as information about data protection, all participants provided written informed consent for participation. In the case of underage participants, the written consent of their legal guardians was obtained. Then, participants were presented with a battery of self-report questionnaires. The study was approved by the local ethics committee.

### Sample

Sample size for the confirmatory factor analysis was calculated in accordance with Kyriazos [34], which resulted in a sample size of at least 320 participants. To prevent dropouts due to careless responding, discontinuation of the study or similar, a sample size of 350 test subjects was aimed for. The final sample consisted of 348 adults with a mean age of 27 years ($SD$ = 10.14, $Min.$ = 14 to Max. = 61). Two-hundred-and-thirty-two respondents (66.66%) identified as female, 111 as male (31.90%), and 5 as diverse (1.44%).

### Instruments

The measurement instrument to be validated in the present study is the German translation [35] of the *Emotion Beliefs Questionnaire* (EBQ) [19], a self-report questionnaire assessing beliefs about the controllability and usefulness of emotions. The total of 16 items (e.g., "Once you feel negative emotions, you can no longer change them.") recorded the controllability and

usefulness of positive and negative emotions. The answers were given on a 7-point response scale, ranging from *1—strongly disagree*, to *4—neither*, to *7—strongly agree*. Higher scores on the scales indicated more maladaptive beliefs about emotions.

**Validation criteria.**   To locate the four facets of the German-language adaptation of the EBQ in a nomological network and assess its convergent and discriminant validity, we investigated its relations to a set of key constructs of emotion processing and perception. Our goal in including this broad range of correlates was to explore the nomological network of emotion beliefs, including measures of emotion regulation and processing, reactivity to and self-efficacy with emotions, and clinical dimensions of, for example, anxiety and depression. We selected certain emotion processing constructs because these constructs were also the focus in Becerra et al.'s [19] original validation study of the EBQ, allowing for direct comparisons of our results.

Further, we expanded the construct validation of the German EBQ by including more independent constructs of emotional processing like sensitivity and reactivity to emotional experiences and perceived self-efficacy in regulating and processing emotions. Lastly, we included clinical measures of emotion processing to allow the first exploration of consequences of differing emotion beliefs in an applied context.

*Emotion regulation*. We measured emotion regulation via the three most prominent existing measures.

The *Difficulties in Emotion Regulation Scale* (DERS) [36] measures difficulties in ER. A five-point Likert scale is used to indicate how frequently the situation presented in an item is experienced by the respondent (i.e., 1—almost never, 0–10%, 2—sometimes, 11–35%, 3—about half the time, 36–65%, 4—most of the time, 66–90%, 5—almost always, 91–100%). The higher the values in the DERS, the more pronounced the difficulties in emotion regulation. The German translation for use with adolescents showed good reliability and validity [37]. In the present study, the short version DERS-18 [38] was used as well as all items of the original scale "Limited access to emotion regulation strategies" (e.g. "When I'm upset, my emotions feel overwhelming.") in order to establish comparability with other studies that used this subscale to measure self-efficacy in emotion regulation.

The *Emotion Regulation Questionnaire* (ERQ) [39], German translation by Abler and Kessler [40], uses ten items to measure the emotion regulation strategies of reappraisal (e.g. "When I want to feel less negative emotion (such as sadness or anger), I change what I'm thinking about.") and suppression (e.g. "I control my emotions by not expressing them.") on a seven-point scale, ranging from *1—not true at all* to *7—completely true*.

The *Perth Emotion Regulation Competency Inventory* (PERCI) [41] uses 32 items to measure the ability to regulate positive (e.g. "I don't know what to do to create pleasant feelings in myself.") and negative (e.g. "When I'm feeling bad, I'm powerless to change how I'm feeling.") emotions. The items are answered on a 7-point Likert scale, *1—strongly disagree*, to *4—neither agree nor disagree*, to *7—strongly agree*, with higher values indicating greater difficulties in emotion regulation.

*Emotional Reactivity*. We assessed the emotional reactivity via two prominent scales.

The *Perth Emotional Reactivity Scale* (PERS) [42] uses 30 items to measure the activation (e.g. "I tend to get upset very easily"), ease (e.g. ""My negative feelings feel very intense) and duration (e.g. "I can remain enthusiastic for quite a while") of positive and negative emotions on a 5-point Likert scale (i.e., 1—very inapplicable, 2—rather inapplicable, 3—neither, 4—rather applicable, 5—very applicable). In the present study, the German translation of the PERS by Schnabel and Witthöft (in preparation) was used.

The *Emotional Reactivity Scale* (ERS) [43] uses 21 items to measure the sensitivity (e.g. "My feelings get hurt easily."), intensity (e.g." When I experience emotions, I feel them very strongly/intensely.") and persistence (e.g. "When something happens that upsets me, it's all I

can think about it for a long time.") of emotional reactivity on a 5-point Likert scale, ranging from *0—does not apply to me at all*, to *1—applies to me a little*, to *2—applies to me to some extent*, to *3—applies to me quite a lot*, to *4—applies to me completely*. Again, for a German translation of the ERS we used the version by Schnabel and Witthöft (in preparation).

*Regulatory Emotional Self-Efficacy*. The *Regulatory Emotional Self-Efficacy* scale (RESE) [44], German version [45], measures perceived self-efficacy in ER. A five-point Likert scale is used to indicate the extent to which the 10 items apply, from *1—not at all good*, to *2—less good*, to *3—fair*, to *4—fairly good*, to *5—very good*. The higher the scores on the RESE, the higher the self-efficacy in emotion regulation. The RESE includes three subscales measuring the perceived self-efficacy in the expression of positive emotions (POS/four items; e.g. "Express joy when good things happen to you?"), in dealing with anger (ANG/three items; e.g. "Avoid flying off the handle when you get angry?") and dependency/stress (DES/three items; e.g. "Keep from getting discouraged in the face of difficulties?"), respectively.

*Depression, anxiety, and stress*. The *Depression Anxiety Stress Scale 21* (DASS-21) [46] is a 21-item-measure of depression (e.g. "1 felt that life was meaningless"), anxiety (e.g. "I felt 1 was close to panic"), and stress (e.g. "I found it hard to wind down") symptoms during the last seven days. The items are rated on a four-point-Likert scale (i.e., from *0—did not apply to me at all*, to *3—applied to me very much or most of the time*).

## Analyses

All analyses were conducted in the statistical language R, version 4.3.2. Our analyses comprised five steps. First, we analyzed the descriptive statistics and zero-order correlations of all 16 items of the German-language adaptation of EBQ. We report the mean, median, standard deviation, minimum and maximum, skewness, and kurtosis, respectively.

Second, we assessed the factorial validity of the EBQ through confirmatory factor analysis (CFA). We computed five different CFA models via maximum likelihood estimation (MLR). One analysis concerned the direct replication of the complete four factor structure of the EBQ assessed in the original paper [19]. The additional four analyses addressed the factorial validity of every single facet theorized by Becerra et al.'s [19] model of emotion beliefs.

Third, we estimated the reliability of the EBQ in terms of internal consistency. We used McDonald's ω complementary to the widely used Cronbach's α because the latter assumes equal factor loadings for all items (i.e., an essentially tau-equivalent measurement model), an assumption that is unlikely to hold for multifactorial models like the EBQ.

Fourth, we report the construct validity of the EBQ by correlating the four facets of emotion beliefs with the aforementioned validation constructs. The aim here was to embed the four emotion belief facets in a nomological network spanning a broad range of conceptually close constructs to investigate divergent and convergent validity.

Fifth and last, to investigate the measurement generalizability of the EBQ, we tested the scale's measurement invariance across gender of respondents and across different levels of emotional reactivity and self-efficacy by means of multi-group CFA [47, 48]. In each measurement invariance analysis, we tested four successive levels of measurement invariance: configural invariance (same measurement model), metric invariance (additionally same loadings), scalar invariance (additionally same intercepts), and strict or uniqueness invariance (additionally same residual variances). To decide on the achieved level of measurement invariance, we relied on conventional cutoffs for changes in fit indices when comparing models with different levels of invariance [49–51]. We tested the measurement invariance of the complete model of four facets based on the tau-congeneric model. We also compared the scale scores (manifest unit-weighed means scores) for each emotion belief facet between genders.

**Table 1. Descriptive statistics of all 16 items of the German translation of the EBQ.**

| Items | M | SD | Min | Max | Skewness | Kurtosis |
|---|---|---|---|---|---|---|
| EBQ-item 1 | 2.70 | 1.37 | 1.00 | 7.00 | .79 | -.09 |
| EBQ-item 2 | 3.27 | 1.67 | 1.00 | 7.00 | .50 | -.78 |
| EBQ-item 3 | 2.52 | 1.47 | 1.00 | 7.00 | 1.18 | .93 |
| EBQ-item 4 | 1.43 | 0.87 | 1.00 | 6.00 | 2.88 | 9.78 |
| EBQ-item 5 | 2.33 | 1.31 | 1.00 | 7.00 | 1.26 | 1.56 |
| EBQ-item 6 | 2.55 | 1.55 | 1.00 | 7.00 | 1.19 | .85 |
| EBQ-item 7 | 1.72 | 1.16 | 1.00 | 7.00 | 2.24 | 5.84 |
| EBQ-item 8 | 1.35 | 0.71 | 1.00 | 6.00 | 2.96 | 11.78 |
| EBQ-item 9 | 2.73 | 1.43 | 1.00 | 6.00 | .80 | -.26 |
| EBQ-item 10 | 2.51 | 1.41 | 1.00 | 7.00 | 1.00 | .57 |
| EBQ-item 11 | 3.10 | 1.71 | 1.00 | 7.00 | .37 | -1.02 |
| EBQ-item 12 | 1.27 | 0.72 | 1.00 | 5.00 | 3.29 | 11.56 |
| EBQ-item 13 | 2.22 | 1.38 | 1.00 | 7.00 | 1.55 | 2.41 |
| EBQ-item 14 | 2.57 | 1.42 | 1.00 | 7.00 | .93 | .34 |
| EBQ-item 15 | 2.64 | 1.55 | 1.00 | 7.00 | .82 | -.04 |
| EBQ-item 16 | 1.42 | 0.92 | 1.00 | 7.00 | 2.86 | 9.41 |

## Results

### Descriptives

We analyzed the descriptive statistics and reference ranges for the German-language adaptation of the EBQ. Table 1 shows the mean (*M*), median (*Mdn*), standard deviation (*SD*), minimum (Min) and maximum (Max), skewness, and kurtosis of all 16 single items of the four facets: Controllability of positive emotions, controllability of negative emotions, usefulness of positive emotions, usefulness of negative emotions. For the inter-item correlations of all 16 single items, see S1 Table in the Supporting information. The intercorrelations between the manifest scores of the four facets of the EBQ in German were moderate to strong, $.30 \leq r \leq .62$ (see S2 Table in the Supporting information).

### Factorial validity

Table 2 shows results for the CFA models for the joint model and for each individual emotion belief facet. Besides inspecting the model Chi-Square test, we consulted the following fit indices to evaluate model fit: comparative fit index (CFI), Tucker-Lewis index (TLI), root mean square error of approximation (RMSEA), and standardized root mean square residual (SRMR).

All five models, in respect to their complexity, showed a satisfying fit according to conventional guidelines (e.g., [52, 53]; but see also against rules of thumb, [54]), after freeing one covariance relationship between item 11 and 15. Due to the complexity of four factors with only 4 items per factor, we deem TLI = .883 acceptable. Although slightly below the conventional cutoff of 0.90, this TLI still reflects a reasonable fit when considering that TLI tends to penalize models with higher complexity more heavily. In this context, the model's complexity —with four factors and relatively few items per factor—creates a more stringent evaluation by the TLI. Nonetheless, the overall fit indices, including the CFI and RMSEA, support that the model adequately captures the present data. Further, the complete emotion beliefs model of the EBQ with all four facets showed a fit very similar to the one reported by Becerra et al. [19] for the English original EBQ. To further support the model's fit, we compared it with the fit of

**Table 2. Confirmatory factor analyses for the whole EBQ in German and each of its four facets.**

| Models | $\chi^2$ | df | p | CFI | TLI | RMSEA (90% C.I.) | SRMR |
|---|---|---|---|---|---|---|---|
| Full model (4 construct factors) | 242.43 | 97 | < .001 | .905 | .883 | .072 [.061;.083] | .063 |
| Alternative model 1 | 638.29 | 103 | < .001 | .651 | .594 | .134 [.124;.144] | .106 |
| Alternative model 2 | 598.43 | 102 | < .001 | .677 | .619 | .129 [.119;.139] | .129 |
| Alternative model 3 | 376.40 | 102 | < .001 | .821 | .790 | .096 [.086;.107] | .100 |
| Alternative model 4 | 352.37 | 100 | < .001 | .836 | .803 | .093 [.083;.104] | .100 |
| Alternative model 5 | 265.87 | 100 | < .001 | .892 | .870 | .075 [.065;.087] | .063 |
| Alternative model 6 | 265.87 | 100 | < .001 | .892 | .870 | .075 [.065;.087] | .063 |
| Controllability (negative emotions) | 4.94 | 2 | .051 | .985 | .956 | .082 [.000;.162] | .031 |
| Controllability (positive emotions) | 0.98 | 2 | .612 | 1.00 | 1.015 | < .001 [.000;.094] | .013 |
| Usefulness (negative emotions) | 0.67 | 2 | .412 | 1.00 | 1.007 | < .001 [.000;.144] | .008 |
| Usefulness (positive emotions) | 5.65 | 2 | .059 | .991 | .972 | .079 [.000;.160] | .021 |

*Note.* Alt. model 1: 1 factor; alt. model 2: 2 valence method factors; alt. model 3: usefulness and controllability factors; alt. model 4: negative and positive controllability factors and one usefulness factor; alt. model 5: negative and positive usefulness factors and one controllability factor; alt. model 6: negative and positive usefulness factors, one controllability factor, and one second-order factor of all three other factors. All models include the one freed covariance relationship between item 11 and 15 to make them fully comparable. Alternative models 5 and 6 show the same indices, indicating that the second-order factor in the latter model does not provide additional explanation of the items' underlying structure.

all six alternative models proposed and tested by Becerra et al. [19]. The four-factor model showed the best fit in this array of alternative models, being significantly better fit to the data than the second best model, $\Delta x^2 = 23.44$, $p < .001$, $\Delta AIC = 17$, $\Delta BIC = 6$).

In addition, we tested the model fit of the four subscales independently, in case researchers want to assess a selected subfactor independently (see for another example, [55]). This can be understood as an extension of why researchers also consider the internal consistency of sub-scales, testing whether items are not only acceptably similar, but also load unidimensional. The four separate models per subscale each showed good fit according to CFI, TLI, SRMR, and RMSEA. We note that a TLI above 1.00 is not uncommon for small item sets and simple models, which subscales tend to be (e.g., [56, 57]), given that it was developed by Tucker and Lewis [58] specifically as a Nonnormed Fit Index (NNFI). S1 Fig in the Supporting information shows the full path model of the four-factor structure of the German EBQ.

## Reliability

Table 3 shows the reliabilities of the four emotion belief facets in the German EBQ. The values of consistency (for the facets and item-corrected) ranged from an acceptable ($\alpha = .68$; $\omega = .70$) to a good internal consistency ($\alpha = .79$; $\omega = .80$).

## Construct validity

Table 4 shows correlations of the emotion regulation subscales of DERS-18 [36, 38], ERQ [39], and PERCI [41], the emotional reactivity subscales of PERS [42] and ERS [43], the facets of

**Table 3. Reliability of the four facets of the EBQ in German.**

| Factors | Internal consistency ($\alpha$) | C.I. (90%) | Internal consistency ($\omega$) | C.I. (90%) |
|---|---|---|---|---|
| Controllability (negative emotions) | .73 | [.68;.77] | .74 | [.68;.80] |
| Controllability (positive emotions) | .68 | [.62;.73] | .70 | [.62;.76] |
| Usefulness (negative emotions) | .72 | [.67;.76] | .75 | [.69;.80] |
| Usefulness (positive emotions) | .79 | [.75;.82] | .80 | [.69;.86] |

**Table 4. Correlations of the facets of the EBQ in German with other individual difference constructs.**

| Variable | Controllability (negative emotions) | Controllability (positive emotions) | Usefulness (negative emotions) | Usefulness (positive emotions) | EBQ total score |
|---|---|---|---|---|---|
| Awareness (DERS) | .12 | .13 | .02 | .03 | .10 |
| Clarity (DERS) | **.47** | **.30** | .23 | **.31** | **.43** |
| Goals (DERS) | **.32** | .17 | .16 | .12 | **.26** |
| Impulse (DERS) | **.33** | **.35** | .19 | .23 | **.36** |
| Nonacceptance (DERS) | **.35** | .25 | **.29** | .23 | .37 |
| Strategies (DERS) | **.44** | **.35** | **.27** | .17 | **.42** |
| Reappraisal (ERQ) | **-.26** | -.22 | -.07 | -.12 | -.22 |
| Suppression (ERQ) | **.26** | .21 | .16 | .16 | **.26** |
| Negative emotion regulation (PERCI) | **.48** | **.38** | **.27** | .19 | **.45** |
| Positive emotion regulation (PERCI) | **.43** | **.37** | .17 | **.35** | **.43** |
| Negative emotion reactivity (PERS) | **.40** | .25 | .17 | .06 | **.31** |
| Positive emotion reactivity (PERS) | -.13 | < .01 | -.01 | -.25 | -.10 |
| Emotion sensitivity (ERS) | **.29** | .18 | .08 | .07 | .21 |
| Emotion intensity (ERS) | **.26** | .22 | .19 | .12 | **.27** |
| Emotion persistence (ERS) | **.35** | .21 | .15 | .07 | **.27** |
| Expression of positive emotions (RESE) | -.14 | -.01 | .09 | -.17 | -.06 |
| Dealing with anger (RESE) | -.22 | -.13 | -.06 | .08 | -.13 |
| Stress/dependency (RESE) | -.17 | -.12 | -.08 | .03 | -.13 |
| Depression (DASS-21) | **.27** | .23 | .12 | .19 | **.26** |
| Anxiety (DASS-21) | **.31** | .24 | .13 | .19 | **.29** |
| Stress (DASS-21) | **.26** | .22 | .14 | .16 | **.26** |
| Vector correlations | **.30** | **.23** | **.15** | **.16** | **.27** |

*Note*. Bold: $p < .001$.

perceived emotional self-efficacy (RESE) [44], and depression, anxiety, and stress symptoms (DASS-21) [46] with the four EBQ-facets.

Closely mirroring recent correlational analyses by Becerra et al. [19], the EBQ facets showed substantial associations with the PERCI facets ($.17 \leq r \leq .48$) and the DASS-21 facets ($.12 \leq r \leq .31$). Further replicating the correlational pattern of the original English EBQ, the two controllability factors of the EBQ in German also show substantially higher associations than the usefulness facets across several validation constructs. Specifically, comparing the factor's average correlations across all validation constructs, we find a descriptive difference between the controllability factor for positive emotions and both usefulness factors and a statistically significant difference for the controllability of negative emotions with both usefulness factors ($p < .05$).

We also present relevant associations of emotion beliefs beyond the original nomological network: Individuals' perceived controllability of emotions is substantially related to their difficulties in regulating their emotions through different channels, most prominently regarding clarity ($.30 \leq r \leq .47$) about and strategies with emotions ($.35 \leq r \leq .44$). We further show that controllability perceptions of emotion are substantially negatively correlated with the suppression of emotions ($-.22 \leq r \leq -.26$) and positively with individuals' ability to reappraise their

emotions ($.21 \leq r \leq .26$). Lastly, emotion controllability is substantially related to how sensitive individuals are to emotional experiences and how intensely and constantly they experience them ($.18 \leq r \leq .35$). General beliefs about the controllability of emotions are neither associated with personal expressions of positive emotions ($-.01 \leq r \leq -.14$), productive ways of dealing with anger ($-.13 \leq r \leq -.22$), nor resistance against stress ($-.12 \leq r \leq -.17$).

Individuals' perception of the usefulness of emotions shows a more nuanced correlational pattern, specifically showing small to moderate associations with emotion regulation measured by PERCI ($.17 \leq r \leq .35$) and the depression, anxiety, and stress subscales of the DASS-21 ($.12 \leq r \leq .19$). We also identify substantial associations between the perceived usefulness of emotions and all facets of regulation difficulties ($.12 \leq r \leq .31$) except with the facet representing individuals' awareness of their emotions ($.02 \leq r \leq .03$).

In conclusion, we replicated the pattern of correlations from Becerra et al.'s [19] original work, with even nuanced correlational differences within the EBQ-construct being reproduced. We also present relevant associations of emotion beliefs beyond the original nomological network, showing that especially controllability perceptions of emotion are substantially associated with emotional self-efficacy and reactivity.

## Measurement invariance

Via multigroup CFA (i.e., using the above tested CFA model for different invariance assumptions of metric, scalar, and strict), we tested the measurement invariance of the German-language translation of the EBQ across genders for the complete emotion belief model and for each of the four facets separately. Results of all five analyses are shown in Table 5.

According to the criteria of Chen [49], Rutkowski and Svetina [51], and Putnick and Bornstein [50], all five tested models reached scalar measurement invariance between the genders (i.e., the factor loadings, intercepts, uniquenesses, and factor variances are equal across genders). For the full model and the three facets controllability of negative emotions, controllability of positive emotions, and usefulness of negative emotions strict invariance can be accepted (i.e., the item residuals are equal across genders). The last facet, namely, usefulness of positive emotions could not reach (partial) strict invariance. Because strict or at least scalar invariance across genders held for all EBQ facets, we can compare the manifest scale scores of the facets across genders. A direct comparison of the composite scores of the four facets of emotion beliefs shows only minor descriptive differences between the two genders, $M_{dif.} = 0.13$, $ps \geq .066$, $r_{r-bi} \leq .12$, namely, slightly lower emotion beliefs in female compared to male respondents. Due to violations of the normality assumption, we used the non-parametric Mann-Whitney U test that indicates effects as rank-biserial correlations ($r_{r-b}$).

For further testing EBQ's measurement robustness, we analyzed the scales' measurement invariance across two different levels (i.e., high vs. low) of emotional reactivity (measured by the composite score of the ERQ) and self-efficacy (measured by the composite score of the RESE), respectively. S4 Table in the Supporting information shows that all EBQ facets show strict measurement invariance across respondents with low vs. high emotional self-efficacy, while S3 Table shows that most EBQ facets show strict invariance across respondents with low vs. high emotional reactivity, with the exception of the controllability of positive emotions that only reached metric measurement invariance. Due to this level of invariance, we can investigate differences in emotion beliefs between low vs. high emotional efficacy and reactivity (through Mann-Whitney U testing due to violations to normality). Highly emotionally reactive individuals showed stronger beliefs in the controllability of negative and positive emotions ($Us = 3043.00$, $ps \leq .040$, $r_{r-bi} \geq .18$) and in the usefulness of negative emotions ($U = 3334.50$, $p = .021$, $r_{r-bi} = .20$), while there was no difference for beliefs in the usefulness of positive

**Table 5. Test of measurement variance across female and male respondents for the whole emotion belief model and its four facets.**

| Models | | Configural | Metric | Scalar | Strict | Partial strict |
|---|---|---|---|---|---|---|
| Complete 4-factor model | $\chi^2$ | 261.35 | 276.97 | 290.18 | 319.86 | |
| | df | 194 | 206 | 218 | 234 | |
| | p | .001 | .001 | .001 | < .001 | |
| | CFI | .929 | .926 | .924 | .910 | |
| | RMSEA (90% C.I.) | .062 [.041;.080] | .061 [.041;.079] | .060 [.040;.078] | .063 [.045;.080] | |
| | SRMR | .071 | .075 | .077 | .082 | |
| | ΔCFI | | .003 | .002 | .014 | |
| | ΔRMSEA | | .001 | .001 | .003 | |
| | ΔSRMR | | .004 | .002 | .005 | |
| | Decision | | Accept | Accept | **Accept** | |
| Controllability (negative emotions) | $\chi^2$ | 7.23 | 8.11 | 12.68 | 16.56 | |
| | df | 4 | 7 | 10 | 14 | |
| | p | .124 | .323 | .242 | .280 | |
| | CFI | .988 | .996 | .990 | .990 | |
| | RMSEA (90% C.I.) | .075 [.000;.161] | .033 [.000;.111] | .043 [.000;.105] | .036 [.000;.092] | |
| | SRMR | .027 | .030 | .038 | .053 | |
| | ΔCFI | | .008 | .006 | < .001 | — |
| | ΔRMSEA | | .042 | .010 | .007 | — |
| | ΔSRMR | | .003 | .008 | .015 | — |
| | Decision | | Accept | Accept | **Accept** | — |
| Controllability (positive emotions) | $\chi^2$ | 2.56 | 3.32 | 10.06 | 13.89 | — |
| | df | 4 | 7 | 10 | 14 | — |
| | p | .633 | .854 | .435 | .458 | — |
| | CFI | 1.000 | 1.000 | .999 | 1.000 | — |
| | RMSEA (90% C.I.) | .000 [.000;.129] | .000 [.000;.070] | .008 [.000;.114] | .000 [.000;.100] | — |
| | SRMR | .022 | .029 | .055 | .068 | — |
| | ΔCFI | | < .001 | .001 | .001 | — |
| | ΔRMSEA | | < .001 | .008 | .008 | — |
| | ΔSRMR | | .007 | .026 | .013 | — |
| | Decision | | Accept | Accept | **Accept** | — |
| Usefulness (negative emotions) | $\chi^2$ | 4.12 | 15.90 | 18.55 | 21.82 | — |
| | df | 2 | 5 | 8 | 12 | — |
| | p | .128 | .007 | .017 | .040 | — |
| | CFI | .993 | .962 | .964 | .966 | — |
| | RMSEA (90% C.I.) | .086 [.000;.205] | .123 [.058;.193] | .096 [.038;.153] | .075 [.016;.125] | — |
| | SRMR | .011 | .051 | .055 | .058 | — |
| | ΔCFI | | .031 | .002 | .002 | — |
| | ΔRMSEA | | .037 | .027 | .021 | — |
| | ΔSRMR | | .040 | .004 | .003 | — |
| | Decision | | Reject | Accept | **Accept** | — |

(*Continued*)

**Table 5.** (Continued)

| Models | | Configural | Metric | Scalar | Strict | Partial strict |
|---|---|---|---|---|---|---|
| Usefulness (positive emotions) | $\chi^2$ | 4.70 | 14.31 | 18.43 | 42.27 | 30.87 |
| | df | 4 | 7 | 10 | 14 | 13 |
| | p | .319 | .046 | .048 | < .001 | .004 |
| | CFI | .998 | .982 | .979 | .930 | .956 |
| | RMSEA (90% C.I.) | .035 [.000;.135] | .085 [.011;.148] | .077 [.007;.131] | .118 [.078;.160] | .098 [.053;.143] |
| | SRMR | .018 | .058 | .065 | .086 | .074 |
| | ΔCFI | | .016 | .002 | .049 | .023 |
| | ΔRMSEA | | .050 | .003 | .041 | .021 |
| | ΔSRMR | | .040 | .007 | .021 | .009 |
| | Decision | | Accept | Accept | Reject | **Reject** |

emotions ($p$ = .124). For self-efficacy, we only found one difference: Individuals that indicate to be emotionally self-efficient report lower beliefs in the controllability of negative emotions ($U$ = 8361.50, $p$ = .002, $r_{r-bi}$ = .24), while we find no differences for the other EBQ-facets ($p$s ≥.103).

## Discussion

The aim of this study was to validate the German EBQ's factorial structure in a large sample and to demonstrate the multi-faceted capacity of the EBQ to assess beliefs about emotions. Our findings corroborate the theoretical four-factor structure in the data and substantiate the high relevance of the EBQ in both research on emotion regulation and in clinical settings.

### Item intercorrelations

The moderate to strong intercorrelations between the manifest scores of the four facets of the EBQ in German showed, that these facets are related but not redundant. Thus, people who believe in the controllability of emotions also believe in the usefulness of emotions. The highest association was between the two controllability subscales, a finding that led Becerra et al. [19] to suggest that distinguishing between the two controllability subscales is unnecessary. In our opinion, this assumption is too strong and leads to loss of information. Therefore, we support the examination of all four subscales.

### Factor structure

The most appropriate solution for our data set was a four-factor model. We were able to replicate the complete four-factor structure proposed by Becerra et al. [19], which also proved to be the best-fitting model in an Iranian and US American [21], a Polish [59], an Italian sample [22], and, most recent, another US American sample [60]. The four factors were also found in the Japanese translation [32], although they found a better fit including two additional second-order factors. In their first study of the EBQ, Becerra et al. [19] claimed that it was unnecessary to distinguish between the two controllability subscales and that doing so created methodological issues that prevented a four-factor solution. Although both controllability subscales were highly correlated in this sample as well, results of the nomological network underscored the importance of conceptualizing each subscale separately. The internal consistencies of all four EBQ subscales were descriptively smaller in this sample compared to the first (English) administration of the EBQ [19], but are still acceptable to good.

## Construct validity

The more individuals believed in the controllability of emotions, the less they used suppression as an emotion regulation strategy. They also reported higher beliefs in their ability to reappraise emotional states, as assessed by the ERQ. Interestingly, while Kashimura et al. [32] did not find a significant relationship between beliefs in emotion controllability and the use of reappraisal, our findings are consistent with Becerra et al. [19], who also found that individuals who believed in the controllability of negative emotions were more engaged in reappraisal. Another interpretation could be that individuals who tend to engage with and reflect on their (negative) emotions, i.e., actively approach their emotions, increasingly realize that emotions are indeed controllable.

Complementing previous research on emotion beliefs, our study is the first to examine the association between emotion reactivity and beliefs about the controllability and usefulness of emotions. Individuals who were more sensitive to emotional experiences, experienced these emotions more intensely, and experienced emotions more consistently than others, also believed less in the controllability of emotions. At the same time, the small associations between the ERS subscales and the EBQ controllability subscales suggest that beliefs about the controllability of negative emotions depend only to some extent on emotional reactivity. Furthermore, this finding adds to the research on the development of emotion beliefs and emotion (dys-)regulation, and indicates that the impact of emotional reactivity on emotion regulation is less than suggested by Nock et al. [43]. From a clinical perspective, cognitive restructuring these uncontrollability beliefs may, in turn, have an impact on pathological levels of emotional reactivity [61–63].

Both EBQ controllability and usefulness beliefs show moderate to high associations with emotion regulation across positive and negative emotions assessed by the PERCI, a finding similar to the original validation study [19], a replication study [60], and those of Ranjbar et al. [21]. Thus, individuals who believed in the controllability and usefulness of positive and negative emotions also reported higher beliefs in their ability to regulate their emotions. Again, the subscale measuring controllability of negative emotions showed the highest correlation.

From a clinical perspective, the relationship between beliefs in the controllability and usefulness of emotions and psychopathology, as assessed by the DASS-21, is of particular interest. In our sample, lower beliefs in the controllability of negative and positive emotions were associated with elevated symptoms of depression, anxiety, and stress, which seems to support the assumptions of learned helplessness [64]. Kashimura et al. [32] found that the General-Controllability composite scale predicted all DASS subscales, whereas the General-Usefulness composite scale did not. Using path analysis, Johnston et al. [65] showed that the Negative-Controllability subscale predicted symptoms of depression, anxiety, and stress, and the Positive-Usefulness scale predicted symptoms of anxiety. The other EBQ subscales (Positive-Controllability and Negative-Usefulness) did not predict any symptoms of depression, anxiety, or stress. Becerra et al. [19] found that all EBQ subscale and composite scores were associated with higher levels of depression, anxiety, and stress symptoms. However, only the General-Controllability composite scale predicted anxiety and stress symptoms in an online sample of adults. In a German sample, the EBQ total score was associated with higher scores on all DASS subscales [20]. In an Australian online sample, controllability beliefs were negatively correlated with psychological distress in both a simple and a multi-predictor model [24]. Ranjbar et al. [21] found that lower beliefs in the controllability and usefulness of emotions were associated with higher levels of depression, anxiety, and stress. In addition, in an Italian sample, individuals who believed in the controllability and usefulness of positive and negative emotions reported lower symptoms of depression, anxiety, and stress. However, there was no

relationship between the DASS-21 Stress subscale and the Positive-Usefulness factor [22]. Finally, the belief that positive emotions are uncontrollable was significantly and negatively associated with anxiety [66]. In summary, previous studies, including the present one, underscore the importance of controllability beliefs in the development and maintenance of psychopathology.

We found that higher emotion regulation difficulties, as assessed by the DERS, were associated with higher beliefs that emotions are both uncontrollable and not useful. In contrast to Rogier et al. [22], we did not find a significant association between perceived usefulness of emotions and individuals' difficulties in being aware of their emotions (DERS awareness subscale). This may be due to the somewhat difficult methodological nature of this subscale in the German translation [37]. When examining the associations between EBQ and DERS at the subscale level, the most prominent relationships were found with the clarity and strategy subscales. Specifically, the EBQ controllability subscale and the DERS strategy subscale could be viewed as proxies for emotion regulation self-efficacy [44, 45], which is an individual's belief in their ability to successfully regulate their emotions. Our study is the first to examine this relationship between EBQ and self-efficacy beliefs. Interestingly, general beliefs about the controllability of emotions (EBQ) were not associated with any personal self-efficacy beliefs about the regulation of specific positive and negative emotions, as assessed with the RESE-D. It could be argued that personal self-efficacy in regulating positive emotions, anger, and stress may be different from the emotions that individuals have in mind when thinking about the controllability of emotions in general.

In summary, individuals who believed in the controllability (and usefulness) of emotions reported stronger beliefs in their ability to regulate their positive and negative emotions, stronger actual emotion regulation and fewer difficulties in doing so, less emotional reactivity, and, in turn, better mental well-being in terms of fewer symptoms of psychopathology.

While the controllability subscales turned out to be more or less homogeneous in terms of several constructs, a different picture emerged when examining the usefulness subscales. In fact, the correlations within the nomological network were smaller than for the controllability subscales. In support of the distinction between beliefs in the usefulness of positive and negative emotions, links between valence-specific beliefs and emotion regulation resulted in theory-confirming associations.

Individuals who believed that negative emotions were not useful (EBQ) tended to be less accepting of their emotional reactions (DERS). In addition to validating the usefulness subscale of the EBQ, this finding has important implications for the development and maintenance of mental health: In two studies, higher acceptance of negative affect was associated with better mental health outcomes and better adaptation to negative stress [4, 67].

Although the controllability subscales tended to show stronger relationships with the constructs examined, such as emotion regulation and symptoms of psychopathology, than the usefulness subscales, significant correlations were also found for the latter. Contrary to the assumption that both beliefs about the controllability and usefulness of emotions are orthogonal [2], these associations were moderate to large in our sample. This seems to indicate that both beliefs are intertwined, and thus may influence further regulatory efforts: Individuals who believe that a particular emotion is not useful may put less effort into regulating it, leading to fewer control experiences and, ultimately, fewer control beliefs about that particular emotion. However, the nature of how belief systems influence each other needs to be examined with longitudinal studies.

Examining measurement invariance, we found individuals with high versus low emotional reactivity, as assessed by the ERS, to differ in beliefs about emotions. Individuals with higher emotional reactivity (higher emotion sensitivity, intensity, and longer emotion persistence;

[43]) were found to hold higher beliefs about controllability of negative and positive emotions, and about the usefulness of negative emotions, than individuals with lower emotional reactivity. These individuals might have more experience with a wide array of emotions and thus might also experience more situations in which (negative) emotions are useful and controllable.

Contrary to the assumption that individuals with high self-efficacy in emotion regulation, as assessed by the RESE-D, would have higher beliefs in the controllability of (negative) emotions, our results point in the opposite direction. Looking at the constructs at item level, one obvious difference between the two scales is the goal of emotion control: While the EBQ focuses on people's emotion beliefs in general, the RESE-D consists of statements about one's own self-efficacy beliefs. This difference in wording, which has implications for emotion beliefs in general [10], may have led individuals with high self-efficacy beliefs to assume that others have less control over their emotion regulation.

### Conceptual advancement and future research

To the best of our knowledge, the present study is the first to validate a German-translated version of the EBQ and to, more generally, comprehensively analyze the validity of the instrument in the context of multiple different facets of emotion regulation, emotional beliefs, psychopathology, and sociodemographic variables. We hasten to especially note the exhaustive exploration of EBQ's measurement invariance not just across sociodemographic characteristics, as is commonly done, but also between different levels of conceptually relevant psychological constructs like emotional reactivity and psychopathological aspects of a respondent's personality.

At the same time, the present survey had its own limitations that need be addressed in future research. A central limitation is the cross-sectional nature of the survey data, which reduces the interpretability of our findings beyond judging statistical fit and correlational patterns. To explore possible causal pathways between emotion beliefs and psychological constructs of reactivity, efficacy, and psychopathology as well as sociodemographic characteristics require longitudinal research in the future. Further, the present sample consisted mainly of young and well-educated adults. To make more general claims about the scale's validities, future research should focus on more diverse samples. Lastly, we investigated EBQ's construct validity in the context of other closely related measures of emotional beliefs, regulation, and reactivity. It may be of particular interest to researchers to comprehensively analyze this vast landscape of measurement instruments that concern constructs that are very closely related. A specific goal may be to find conceptual overlap among the vast array of instruments and to condense them into overarching, central dimensions that capture beliefs about, regulation of, and reactivity to emotions.

### Practical implications

In psychotherapy, challenging and transforming existing maladaptive beliefs about the nature of emotions is a promising strategy to provide belief and for productive personal change. In accordance with this, the associations found in this study suggest that beliefs in the controllability and usefulness of positive and negative emotions are associated with better mental health outcomes in terms of fewer difficulties with emotion regulation in general, better abilities to regulate positive and negative emotions, and fewer symptoms of psychopathology. Addressing the reverse causal path, focusing on increasing a patient's acceptance of different emotions and establishing the use of emotion regulation strategies can also improve their beliefs in the controllability and usefulness of emotions, which may, in turn, ensure a healthy way of dealing with one's own emotions.

## Conclusion

The modulation of positive and negative emotions based on goals, is a crucial function for a person's well-being and general functioning. Factors influencing successful emotion regulation include beliefs about emotions, such as the controllability and usefulness of emotions. The Emotion Beliefs Questionnaire (EBQ) was developed to assess these beliefs and has shown promise in predicting emotion regulation and psychopathology across different countries. The present paper has shown that the scale's validity is also supported in a German sample, while also expanding the nomological network further. Measuring the defining aspects of emotion regulation reliably and accurately seems to be imperative to support individuals to deal with problematic or overwhelming situations in everyday life or more clinical situations. Future research should focus on further expanding and understanding the causal relationships between different aspects of emotion regulation, such as beliefs, abilities to regulate, and reactivity or sensitivity to emotional situations.

## Supporting information

**S1 Table. Inter-item correlations of EBQ items.**
(PDF)

**S2 Table. Intercorrelations between the four EBQ facets.**
(PDF)

**S3 Table. EBQ measurement invariance across emotional reactivity.**
(PDF)

**S4 Table. EBQ measurement invariance across emotional self-efficacy.**
(PDF)

**S1 Fig. Path model of the four-factor EBQ structure.**
(PDF)

## Acknowledgments

We would like to express our gratitude to Paula Vogt for her invaluable assistance in recruiting the sample.

## Author Contributions

**Conceptualization:** Raphael Gutzweiler, David J. Grüning.

**Formal analysis:** David J. Grüning.

**Methodology:** David J. Grüning.

**Project administration:** Raphael Gutzweiler.

**Resources:** Raphael Gutzweiler.

**Writing – original draft:** Raphael Gutzweiler, David J. Grüning.

**Writing – review & editing:** Raphael Gutzweiler, David J. Grüning.

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
