## [Decision Letter · Decision Letter 0]

12 Sep 2024

PONE-D-24-31699Measuring four facets of emotion beliefs in Germany: A German-language adaptation of the EBQ and its comparability across gender and different emotion abilitiesPLOS ONE

Dear Dr. Gutzweiler,

Thank you for submitting your manuscript to PLOS ONE. After careful consideration, we feel that it has merit but does not fully meet PLOS ONE’s publication criteria as it currently stands. Therefore, we invite you to submit a revised version of the manuscript that addresses the points raised during the review process.

We look forward to receiving your revised manuscript.

Kind regards,

Henri Tilga, PhD

Academic Editor

PLOS ONE

**Journal Requirements:**

Reviewers' comments:

Reviewer's Responses to Questions

**Comments to the Author**

1. Is the manuscript technically sound, and do the data support the conclusions?

Reviewer #1: Yes

Reviewer #2: Yes

2. Has the statistical analysis been performed appropriately and rigorously? 

Reviewer #1: Yes

Reviewer #2: Yes

3. Have the authors made all data underlying the findings in their manuscript fully available?

Reviewer #1: No

Reviewer #2: Yes

4. Is the manuscript presented in an intelligible fashion and written in standard English?

Reviewer #1: Yes

Reviewer #2: Yes

5. Review Comments to the Author

**Reviewer #1: **Thank you for giving me the opportunity to review this study in the field of emotion regulation.

This study presents alot of valuable work. However, it needs some alterations before it gets published. From my perspective, there is some work to be done and alot of alterations which although many are managable...

INTRODUCTION

Please, start with a brief review linking otgether all of the variables used in the study.

Do not use stats in the Introduction and Discussion sections.

Keep a consistent way in the presentation of findings of previous studies- think of the most appropriate way to present previous studies and stick to it (e.g. refer to specific coefficients, refer to the problematic ones, refer to validity or/and reliability coefficients and any other relevant coefficients…).

The aim of the study to be rewritten in a clear and straightforward way. Mention the contribution of the present study as (a), (b), (c), (d)… after reporting the limitations of previous studies.

Factorial Validity: Give more information about “we deem TLI = .883 acceptable”.

Measurement invariance: Start this section with the test you use

Discussion: the first paragraph should give the strong findings of the present study. Then you can discuss the findings of your stats analysis in a story telling way.

Each section should start with your findings and then discuss them in relation to other studies e.g. Becerra et al …

Check the use of the APA across the document, the tables as well…

**Reviewer #2:** In the introduction (L. 105 to 108), the authors summarised the studies validating the EBQ. They neglected to include this study: Kashimura, M., Ishizu, K., & Becerra, R. (2023). Psychometric Examination of the Japanese Version of the Emotion Beliefs Questionnaire 1 2 3 4. Japanese Psychological Research. (Ok, I see now this paper was been brought up in the discussion)Change "...of the below-depicted study" (L.152) to "...of the present study"

At the end of the introduction (L. 153 to 155), the authors state "By using multiple measures to validate the EBQ, we hope to show a broader picture of the EBQ and its abilities to measure beliefs about emotions". Don't say "we hope", just state that you will show. Also, I'm not sure what a "broader picture is". Operationalise this more precisely.

L175 "(EBQ; (18); German translation: (34))", check this journal and APA guidelines for the use of brackets within brackets.

L. 396, change from "For self-efficacy, we only find one difference:..." to "...found..." (past tense).

L. 406 to 409 "a finding that led Becerra et al. (18) to suggest that distinguishing between the two controllability subscales is unnecessary. In our opinion, this assumption is too strong and leads to loss of information. Therefore, we support the examination of all four subscales". Good finding and I agree with the authors' interpretation.

L. 460. "The other EBQ subscales did not predict any symptoms". In the context of this section, I'm not sure what this sentence refers to.

Sometimes the subscale "General controllability" is referred to as "General Controllability" and other times as "General Control". Make this consistent.

L. 517. "...we found individual with..." change to "...individuals..."

Good job!

6. PLOS authors have the option to publish the peer review history of their article (what does this mean?). If published, this will include your full peer review and any attached files.

Reviewer #1: No

Reviewer #2: No

---

## [Author Response · Author response to Decision Letter 0]

16 Oct 2024

Response to Reviewers

We thank the reviewers for their helpful comments which allows a higher quality of the manuscript (PONE-D-24-31699). Please find enclosed the revision of our submission to PLOS ONE. We have carefully considered each comment and have implemented the suggested changes or responded to comments. In the following sections, we respond on a point-by-point basis to the issues raised by the reviewers. We hope that we have dealt satisfactorily with all concerns.

Reviewer #1: Thank you for giving me the opportunity to review this study in the field of emotion regulation.

This study presents a lot of valuable work. However, it needs some alterations before it gets published. From my perspective, there is some work to be done and a lot of alterations which although many are managable...

INTRODUCTION

Please, start with a brief review linking otgether all of the variables used in the study.

We included a brief review linking all relevant variables used in this study.

Do not use stats in the Introduction and Discussion sections.

We removed the stats in the Introduction and Discussion section.

Keep a consistent way in the presentation of findings of previous studies- think of the most appropriate way to present previous studies and stick to it (e.g. refer to specific coefficients, refer to the problematic ones, refer to validity or/and reliability coefficients and any other relevant coefficients…).

We adjusted parts of the Introduction section.

The aim of the study to be rewritten in a clear and straightforward way. Mention the contribution of the present study as (a), (b), (c), (d)… after reporting the limitations of previous studies.

We adjusted our presentation of the aims of the study.

Factorial Validity: Give more information about “we deem TLI = .883 acceptable”.

We added following more information about this in L 322-327. At this point, TLI is sensitive to overfit, which is due to complex models that run the risk to overfit the present data. But it is far from any negative value like below .800.

Measurement invariance: Start this section with the test you use

We included additional information in L 392-393.

Discussion: the first paragraph should give the strong findings of the present study. Then you can discuss the findings of your stats analysis in a story telling way.

Each section should start with your findings and then discuss them in relation to other studies e.g. Becerra et al …

We included a short introduction of the strong findings in the Discussion and adjusted the way, we discussed the findings.

Check the use of the APA across the document, the tables as well…

We have checked the document (and tables) for deviations from APA, however, we could not find any substantial ones. Would you mind pointing us to specific lines so that we can adjust the manuscript?

Reviewer #2: In the introduction (L. 105 to 108), the authors summarised the studies validating the EBQ. They neglected to include this study: Kashimura, M., Ishizu, K., & Becerra, R. (2023). Psychometric Examination of the Japanese Version of the Emotion Beliefs Questionnaire 1 2 3 4. Japanese Psychological Research. (Ok, I see now this paper was been brought up in the discussion)Change "...of the below-depicted study" (L.152) to "...of the present study"

We now included the study by Kashimura et al. (2023) in the introduction. 

At the end of the introduction (L. 153 to 155), the authors state "By using multiple measures to validate the EBQ, we hope to show a broader picture of the EBQ and its abilities to measure beliefs about emotions". Don't say "we hope", just state that you will show. Also, I'm not sure what a "broader picture is". Operationalise this more precisely.

We rewrote this section (L. 170).

L175 "(EBQ; (18); German translation: (34))", check this journal and APA guidelines for the use of brackets within brackets.

We rewrote this section.

L. 396, change from "For self-efficacy, we only find one difference:..." to "...found..." (past tense).

We adjusted the expression.

L. 406 to 409 "a finding that led Becerra et al. (18) to suggest that distinguishing between the two controllability subscales is unnecessary. In our opinion, this assumption is too strong and leads to loss of information. Therefore, we support the examination of all four subscales". Good finding and I agree with the authors' interpretation.

Thank you!

L. 460. "The other EBQ subscales did not predict any symptoms". In the context of this section, I'm not sure what this sentence refers to.

We described these finding in more detail (L. 499).

Sometimes the subscale "General controllability" is referred to as "General Controllability" and other times as "General Control". Make this consistent.

We adjusted the expression throughout the manuscript.

L. 517. "...we found individual with..." change to "...individuals..."

We adjusted the expression.

Good job!

---

## [Decision Letter · Decision Letter 1]

31 Oct 2024

PONE-D-24-31699R1Measuring four facets of emotion beliefs in Germany: A German-language adaptation of the EBQ and its comparability across gender and different emotion abilitiesPLOS ONE

Dear Dr. Gutzweiler,

Thank you for submitting your manuscript to PLOS ONE. After careful consideration, we feel that it has merit but does not fully meet PLOS ONE’s publication criteria as it currently stands. Therefore, we invite you to submit a revised version of the manuscript that addresses the points raised during the review process.

We look forward to receiving your revised manuscript.

Kind regards,

Henri Tilga, PhD

Academic Editor

PLOS ONE

Reviewers' comments:

Reviewer's Responses to Questions

**Comments to the Author**

1. If the authors have adequately addressed your comments raised in a previous round of review and you feel that this manuscript is now acceptable for publication, you may indicate that here to bypass the “Comments to the Author” section, enter your conflict of interest statement in the “Confidential to Editor” section, and submit your "Accept" recommendation.

Reviewer #1: All comments have been addressed

Reviewer #3: (No Response)

2. Is the manuscript technically sound, and do the data support the conclusions?

Reviewer #1: Yes

Reviewer #3: Partly

3. Has the statistical analysis been performed appropriately and rigorously? 

Reviewer #1: Yes

Reviewer #3: No

4. Have the authors made all data underlying the findings in their manuscript fully available?

Reviewer #1: Yes

Reviewer #3: No

5. Is the manuscript presented in an intelligible fashion and written in standard English?

Reviewer #1: Yes

Reviewer #3: Yes

6. Review Comments to the Author

Reviewer #1: The authors have taken account of all of my comments and thy seem to have throughtfully revised the paper.

Reviewer #3: It seems that adolescents (min. age was 14) participated in this study. Did you receive the inform consent from parents in such cases?

Statistical programs and their versions with corresponding analyses should be indicated.

Table 2: TLI values in some cases are above 1, and RMSEA values are 0. Please explain such values. Overall, what is the reason of testing separate subscales with CFA? It is uncommon practice. This is a sign of model misspecification error.

Please test all the models used in Becerra et al. Please examine whether the 4-factor solution is the best one across all possible solutions presented in the original and other validation studies.

I feel the authors can extensively reconsider their paper based on these comments. In the current form, the analyses conducted are definitely insufficient, and even misleading in some cased. For instance, the sentence "The most appropriate solution for our data set was a four-factor model." suggests that other models were tested. It is not truth, as only the 4-factor model was tested. Testing separate subscales with CFA could not be considered as testing of the whole questionnaire (moreover, such practices are uncommon and should be justified).

7. PLOS authors have the option to publish the peer review history of their article (what does this mean?). If published, this will include your full peer review and any attached files.

Reviewer #1: **Yes: **Prof. Evangelia Karagiannopoulou

University of Ioannina, Greece

Honorary Professor University College London, UCL, UK

Reviewer #3: No

---

## [Author Response · Author response to Decision Letter 1]

2 Dec 2024

Reviewer 1

1. The authors have taken account of all of my comments and thy seem to have throughtfully revised the paper.

We very much appreciate this evaluation and thank the reviewer again for their insightful comments in the review process. 

Reviewer 3

1. It seems that adolescents (min. age was 14) participated in this study. Did you receive the inform consent from parents in such cases?

Yes we did, as mentioned in lines 177-178: “In the case of underage participants, the

written consent of their legal guardians was obtained.”

2. Statistical programs and their versions with corresponding analyses should be indicated.

We have now included the statistical program used (i.e., R), its version, and indicated that all analyses were done in R (p. 12): “All analyses were conducted in the statistical language R, version 4.3.2.”

3. Table 2: TLI values in some cases are above 1, and RMSEA values are 0. Please explain such values.

For more complex models TLIs > 1.00 can point to misspecification and/or overfitting. As it is occurring here, it’s not uncommon, though, for small item sets with simple (i.e., one-factor) models like the structure we usually encounter for subscales exactly as the ones presented in the present mansucript. We now added a note on this in the revised manuscript (p. 16): 

“We note that a TLI above 1.00 is not uncommon for small item sets and simple models, which subscales tend to be (e.g., Goretzko et al., 2023; Schermelleh-Engel et al., 2003). Given that it was developed by Tucker and Lewis (1973) specifically as a Nonnormed Fit Index (NNFI), values above 1.00 are not implausible.”

We thank the reviewer for noting the indication of RMSEA being exactly 0. We revised this to now indicate RMSEA < .001, as the R-analysis has indicated it as 0 due to simplifying after three decimal points.

4. Overall, what is the reason of testing separate subscales with CFA? It is uncommon practice.

The idea behind testing the fit of independent factors besides testing the whole model is to test if the different factors fall apart if they are not contrasted with the other factors in one model (i.e., dissimilarity in responses to the other items). To use the different subscales of the model independently, their model fit has to be evaluated independently. We have added a note on this to the revised manuscript (p. 15 to 16): “In addition, we tested the model fit of the four subscales independently, in case researchers want to assess a selected subfactor independently (see for another example, Grüning & Lechner, 2022). This can be understood as an extension of why researchers also consider the internal consistency of subscales, testing whether items are not only acceptably similar, but also load unidimensional.”

In this paragraph, we now also revised “factors” to “subscales” to further make clear what the intention of single subscale model fit testing is.

5. Please test all the models used in Becerra et al. Please examine whether the 4-factor solution is the best one across all possible solutions presented in the original and other validation studies.

We have now also tested all alternative models by Becerra et al. (2018) showing that the four-factor model shows the best fit across all of them. Thank you very much for this critical suggestion which added substantial support to the proposed four-factor model of EBQ. We have added the results of these analyses to Table 2 and commented on the comparisons in the revised manuscript (p. 15): "To further support the model’s fit, we compared it with the fit of all six alternative models proposed and tested by Becerra et al. (18). The four-factor model showed the best fit in this array of alternative models, fitting the data significantly better than the second best model, Δx2 = 23.44, p < .001, ΔAIC = 17, ΔBIC = 6).”

---

## [Decision Letter · Decision Letter 2]

5 Dec 2024

Measuring four facets of emotion beliefs in Germany: A German-language adaptation of the EBQ and its comparability across gender and different emotion abilities

PONE-D-24-31699R2

Dear Dr. Gutzweiler,

We’re pleased to inform you that your manuscript has been judged scientifically suitable for publication and will be formally accepted for publication once it meets all outstanding technical requirements.

Kind regards,

Henri Tilga, PhD

Academic Editor

PLOS ONE

Additional Editor Comments (optional):

Reviewers' comments:

Reviewer's Responses to Questions

**Comments to the Author**

1. If the authors have adequately addressed your comments raised in a previous round of review and you feel that this manuscript is now acceptable for publication, you may indicate that here to bypass the “Comments to the Author” section, enter your conflict of interest statement in the “Confidential to Editor” section, and submit your "Accept" recommendation.

Reviewer #3: (No Response)

2. Is the manuscript technically sound, and do the data support the conclusions?

Reviewer #3: Yes

3. Has the statistical analysis been performed appropriately and rigorously? 

Reviewer #3: Yes

4. Have the authors made all data underlying the findings in their manuscript fully available?

Reviewer #3: Yes

5. Is the manuscript presented in an intelligible fashion and written in standard English?

Reviewer #3: Yes

6. Review Comments to the Author

Reviewer #3: The authors must not ignore the previous comments in the reviewer's review report. The authors did not address this comment: "I feel the authors can extensively reconsider their paper based on these comments. In the current form, the analyses conducted are definitely insufficient, and even misleading in some cased. For instance, the sentence "The most appropriate solution for our data set was a four-factor model." suggests that other models were tested. It is not truth, as only the 4-factor model was tested. Testing separate subscales with CFA could not be considered as testing of the whole questionnaire (moreover, such practices are uncommon and should be justified).".

Despite this fact, the paper can be accepted.

7. PLOS authors have the option to publish the peer review history of their article (what does this mean?). If published, this will include your full peer review and any attached files.

Reviewer #3: No

---

## [Editor Report · Acceptance letter]

9 Dec 2024

PONE-D-24-31699R2 

PLOS ONE

Dear Dr. Gutzweiler, 

I'm pleased to inform you that your manuscript has been deemed suitable for publication in PLOS ONE. Congratulations! Your manuscript is now being handed over to our production team.

Kind regards, 

on behalf of

Dr. Henri Tilga 

Academic Editor

PLOS ONE